# Influence of CLOCK Gene Variants on Weight Response after Bariatric Surgery

**DOI:** 10.3390/nu14173472

**Published:** 2022-08-24

**Authors:** Macarena Torrego-Ellacuría, Ana Barabash, Pilar Matía-Martín, Andrés Sánchez-Pernaute, Antonio J. Torres, Alfonso L. Calle-Pascual, Miguel A. Rubio-Herrera

**Affiliations:** 1Department of Endocrinology and Nutrition, Hospital Clínico San Carlos, IdISSC, 28040 Madrid, Spain; 2Department of Medicine, Faculty of Medicine, Universidad Complutense, 28040 Madrid, Spain; 3Centro de Investigación Biomédica en Red de Diabetes y Enfermedades Metabólicas Asociadas (CIBERDEM), 28029 Madrid, Spain; 4Department of Surgery, Hospital Clínico San Carlos, IdISSC, 28040 Madrid, Spain

**Keywords:** CLOCK gene, obesity, bariatric surgery, SNP, weight regain

## Abstract

The Circadian Locomotor Output Cycles Kaput (CLOCK) gene has been linked to metabolic dysfunction and obesity. The purpose of this study was to analyze the association between single nucleotide polymorphisms (SNPs) of CLOCK gene with obesity and with long-term weight response after different bariatric surgery (BS) techniques. The cohort includes 375 patients with morbid obesity (MO) and 230 controls. In the association study of SNPs with weight response we combined several variables as phenotype at 6 years after surgery. The study protocol was registered in ISRCTN (ID80961259). The analysis of the selected SNPs was performed by allelic discrimination using Taqman^®^ probes. The genotype association study was performed using the SNPStats program, with comparisons adjusted for sex, age, initial Body Mass Index, type 2 diabetes and hypertension diagnosis, and type of surgery. In the case-control study two of three SNPs were significantly associated with MO. The variant rs1801260 had a protective effect for MO whereas the TT genotype of rs3749474 variant had the strongest association with MO (OR = 2.25 (1.39–3.66); *p* = 0.0006). In the linear regression analysis both variants showed significant association with long-term weight loss and weight regain after BS, independently of the pre-surgery patient profile.

## 1. Introduction

Studies in animal models [1] and in humans [2,3,4] have shown associations between chronobiology and the prevalence of obesity, highlighting the involvement of the Circadian Locomotor Output Cycles Kaput (CLOCK) factor in metabolic dysfunction [5]. Shift work, sleep deprivation, exposure to bright artificial light at night [5], and timing of food intake [6] have been associated with increased adiposity. Alterations in circadian rhythm in stomach and jejunum explants in morbid obesity (MO) subjects have also been observed [7]. 

The single nucleotide polymorphisms (SNP) variants rs3749474 and rs1801260 have been described in association with obesity and weight loss following dietary and behavioral weight loss intervention [8]. These two polymorphisms have also been described in the interrelationship between dietary intake and weight loss [9,10].

In MO subjects undergoing bariatric surgery (BS), an association between an evening chronotype and lower long term weight loss after intervention has been described [4]. Moreover, timing of main meal intake and weight loss effectiveness was also studied, showing lower weight loss in subjects who had a later main meal intake [11]. However, the role of CLOCK variants after BS has not been extensively studied and the association with long-term weight regain (WR) has not been previously described. 

The aim of this study was to evaluate the association between CLOCK gene polymorphisms with MO diagnosis and with weight loss and its long-term maintenance after different BS surgical techniques. 

## 2. Materials and Methods

This was a single-center retrospective study based on a prospective database from Hospital Clínico San Carlos, Madrid (HCSC). It included 375 cases, patients undergoing BS (BMI > 40 kg/m^2^ or BMI ≥ 35 kg/m^2^ associated with comorbidity) and 230 controls (healthy subjects without comorbidities with BMI 18.5–25 kg/m^2^). In both cohorts, subjects were aged between 18 and 65 years and all were Caucasian. The cases were selected from the 510 subjects who underwent a first bariatric surgical procedure between 2009 and 2014, after applying the exclusion criteria previously reported [12] together with the exclusion of patients of Latin ethnicity and without genotyping. The project was approved by the HCSC Clinical Research Ethics Committee (16 February 2009). The study protocol was registered at https://www.isrctn.com/ (accessed on 4 August 2022) (ID ISRCTN80961259). 

The genetic variants included in the study were SNP-like variants of CLOCK gene: rs3749474, rs1801260, and rs4580704. The case-control study analyzed the association of the 3 SNPs with the MO condition. The association between SNPs and weight response after BS was analyzed in the case cohort. Demographic, clinical, and anthropometric information was collected in the electronic health records prior to surgery. Type 2 diabetes (T2D) and hypertension (HTN) were diagnosed and categorized. The different surgical techniques (ST) were sleeve gastrectomy (SG), Roux-en-Y gastric bypass (RYGB), biliopancreatic diversion with or without duodenal switch (BPD-DS), and single anastomosis duodeno-ileal bypass with sleeve gastrectomy (SADI-S), as malabsorptive procedures. The type of surgical technique was chosen according to clinical practice criteria of hospital protocol, based on age, BMI, and comorbidities. 

The 375 cases were followed up after surgery, with annual appointments up to 8 years with weight measurements [12]. The main variables for assessing weight response include percentage of total weight lost (%TWL); excess weight loss (%EWL), with ideal weight calculated for a BMI of 25 kg/m^2^; and WR as percentage of maximum weight loss (%WR_MWL) [13]. Nadir weight was determined based on all the postoperative weight measures available, considering the lowest value. The end of clinical follow-up was established in year 6 since it was the common period of follow-up of the entire sample according to the inclusion dates. 

### 2.1. DNA Extraction and Genotyping of Samples 

Two peripheral blood tubes (EDTA) of 10 mL each were collected from each patient. DNA was extracted from peripheral leukocytes after a series of washes with red cell lysis buffer and further treated with the DNAzol^®^—Genomic DNA Isolation Reagent extraction kit, following the manufacturers protocol. Concentration and purity were determined using a NanoDrop™ 2000/2000C spectrophotometer. Genotyping was performed using predesigned TaqMan assays for rs3749474 (C__26405955_10), rs1801260 (C___8746719_20), and rs4580704 (C__28028791_10) using a 7500 Fast Real-Time PCR System (Applied Biosystems, Foster City, CA, USA). A genotyping call rate over 95% per plate, negative sample controls, and three well-differentiated genotyping clusters were required to validate results.

### 2.2. Statistical Analyses

Data were expressed as the mean and standard deviation or median and interquartile range for continuous variables and absolute and relative frequencies for categorical variables. The SNPStats software was used to evaluate Hardy–Weinberg equilibrium and the genotype association study (SNPs-MO condition and SNPs-weight loss), under multiple inheritance models with respect to allele reference homozygotes [14]: co-dominant, dominant, recessive, over-dominant, and log-additive [15]. A linear regression analysis was performed for quantitative response variables (%TWL, %EWL, %WR_MWL), expressing the results with the mean, standard error, and mean differences (95%CI). For response variables coded as a binary variable (MO diagnosis, %TWL > 20% [16], %EWL > 50% [17]) a logistic regression analysis was performed, expressing the results including genotype frequencies, proportions, and OR (95%CI). 

The case-control study was adjusted by sex. The association study of SNPs and weight loss and regain after BS was performed on the overall sample and stratified by type of surgery: non-malabsorptive techniques (SG and RYGB) vs. malabsorptive techniques. Comparisons were adjusted for sex, age, initial BMI, pre-surgery T2D and HTN diagnosis, and type of surgery. All *p* values lower than 0.05 were deemed statistically significant.

## 3. Results

### 3.1. Case Control Association Study 

The case-control association study included 605 subjects (375 cases and 230 controls) and the 3 SNPs of CLOCK gene. The mean BMI was 21.72 ± 1.92 kg/m^2^ in the controls and 44.94 ± 6.88 kg/m^2^ in the MO category. Regarding sex distribution, both groups were mostly composed of women, representing 73.7% in the controls and 71.1% in the cases. Table 1 lists the SNPs included, with genotype absolute and relative frequency in controls and cases. The rs3749474 variant of the CLOCK gene in homozygosis (TT) presented the highest association with MO (OR (95%CI) = 2.25 (1.39–3.66); *p* = 0.0006). The rs1801260 variant had a protective effect for MO (OR = 0.39 (0.20–0.74); *p* = 0.004) according to recessive inheritance model. The rs4580704 variant showed no association with MO diagnosis.

### 3.2. Association Study with Weight Response

The surgical techniques performed in the cases were 16% SG, 54.66% RYGB, and 29.3% malabsorptive (77% SADI-S, 23% BPD-DS), with a median follow-up of 6 years (IQR = 5–8) after BS. Table 2 shows the description of the patient profile in the case cohort, including the pre-surgery variables and weight loss and weight regain variables included as phenotypes in the study of the association of genetic variants with weight response after BS. 

The rs1801260 and rs3749474 variants showed association with weight response after BS after adjusting for age, sex, T2D, HTN, type of surgical technique, and initial BMI. Figure 1 and Figure 2 illustrate the mean values of %TWL between genotypes throughout follow-up. Mean values of weight response variables stratified by genotypes of the 3 SNPs included are shown in Table 3. The rs1801260 variant showed association in the global cohort at year 6 following an additive model, with a significant mean difference in %TWL (Mean difference (IC95%) = 1.85 (0.07–3.62); *p* = 0.042) and in %WR_MWL (Mean difference (IC95%) = −3.27 (−6.42–−0.12); *p* = 0.042). The rs3749474 and the rs4580704 variant showed no significant association with weight response at year 6 in the global cohort. No statistically significant differences were found for the %TWL achieved at nadir nor for the %EWL variable or %TWL and %EWL variables coded according to the cut-off points of 20% and 50%, respectively, at the end of follow-up with any of the 3 SNPs. 

In the cohort of malabsorptive techniques (N = 110), an association was observed between the rs3749474 variant of the CLOCK gene and weight loss and regain at the end of the follow-up. Carriers of the TT genotype achieved a significantly lower %TWL (Mean difference (IC95%) = −5.80 (−10.57–−1.03); *p* = 0.019) and higher %WR_MWL (Mean difference (IC95%) = 8.41 (1.07–16.00); *p* = 0.027), Figure 3. Carriers of the T allele were also associated with the EWL > 50% (OR = 0.39 (0.20–0.74); *p* = 0.004), following a dominant model. The rs1801260 and the rs4580704 variants showed no significant association with weight response when stratified by surgical techniques.

## 4. Discussion

The knowledge generated by large-scale meta-analyses of data from genome-wide association studies (GWAS) has identified more than 150 common genetic variants strongly correlated with obesity or adiposity traits, although the individual impact of each variant is modest [18,19,20]. The parameter most commonly used as a phenotype to evaluate the degree of obesity is the BMI, due to its good correlation with the percentage of body fat [21]. Very few studies include patients with grades III-IV obesity, so the characteristics of the subjects in our cohort, including MO cases together with a sample of controls of the same demographic profile and ethnicity without obesity, are of great interest in the identification of genetic markers associated with the pathophysiology of MO.

In our work we have observed the association between CLOCK gene polymorphisms and obesity previously described in other studies, where carriers of the T allele of the SNP rs3749474 showed a significantly higher degree of obesity according to BMI and abdominal obesity than carriers of the other genotypes [8]. These results were replicated in our sample, where carriers of the T allele in homozygosis presented the greatest association with MO, with a 1.39–3.66 increased risk, regardless of sex. The allele and genotype frequency in the controls was comparable to that described in the European population database [14].

There is currently little evidence regarding underlying mechanisms between circadian physiological rhythms and increased risk of metabolism disorders [22]. In a recent work, Espinosa-Salinas et al. studied the interrelationship between chronobiological aspects controlled by the CLOCK gene and their influence on the incidence of obesity, with the aim of identifying interactions between the CLOCK gene variants and the regulation of appetite in a sample of 442 subjects aged 18 to 65 years [23]. A significant influence was found in terms of the effects of appetite on waist circumference with respect to the rs3749474 variant (*p* < 0.001), where carriers of the risk allele increased waist circumference by about 14 cm for each increase in appetite level. These results suggest that the effects of appetite on waist circumference may be partially modulated by the rs3749474 variant. In another study evaluating the distribution of energy intake and macronutrients consumption, and how its effect on nutritional status can be modulated by the presence of the rs3749474, a significant interaction was observed between this variant and the circadian evening carbohydrate intake impact on a higher BMI [9]. In the study by Loria-Kohen et al., participants carrying CLOCK rs3749474 T allele showed a positive association between the change in percentage intake of dietary fat and change in BMI [10]. 

In relation to the association of CLOCK gene variants and weight loss phenotypes, little research has been done in this area, with very few studies involving patients undergoing BS. In the dietary intervention studies of Garaulet et al. in subjects with obesity, an association was observed between the rs1801260 variant and a greater degree of obesity, as well as a greater resistance to weight loss in response to a low-energy diet [8]. It has also been described a gene diet interaction of rs1801260 in low-fat diet compared with Mediterranean diet and glucose metabolism [24]. In our study, carriers of the A allele of the rs1801260 variant of the CLOCK gene show a greater degree of obesity and significant lower weight loss and higher weight regain in the long term, regardless of the pre-surgery patient profile. This variant has also been described in the study by Ruiz Lozano et al., associated with a lower efficacy of weight loss after BS [4]. In the cohort of malabsorptive techniques, the rs3749474 variant of the CLOCK gene was associated with weight response, and carriers of the T allele had a lower weight loss at the end of the follow-up and a significant greater weight regain. To our knowledge this is the first study demonstrating an association between the SNP rs3749474 and long-term weight response in malabsorptive techniques. 

Previous data by our group, based on the original cohort, identified potential predictors of long-term weight regain [12], included among the adjustment variables used in the regression analysis. Significant association results are independent of the patient profile before undergoing BS and the type of surgical technique undergone. In the absence of a standardized criterion to evaluate weight response after BS [25], we have combined several variables as phenotype in the association study. At year 6 the magnitudes achieved of weight loss were similar or even superior to previous studies [26,27,28] that include a long-term follow-up [25]. In addition, weight regain measured with respect to maximum weight loss achieved was lower than previous studies [13,29]. Despite achieving a homogeneous and high magnitude weight loss, without great variability between the cohort, differences in phenotype can be observed when stratifying by genetic variants.

Limitations related to long-term response to BS include the failure to collect information on dietary intake and physical activity, variables with potential impact on weight response. Another issue is that patients were not randomized from baseline for surgical procedure selection; however, the adjustments made in the association analysis minimize the lack of randomization in surgical technique assignment. Despite these limitations, the strengths include a high retention rate of cases in a long-term follow-up study, including three modalities of BS techniques, especially malabsorptive techniques in which there are few genetic association studies, and the exclusion of Latino subjects in the cohort of cases, which enhances homogeneity and quality. 

In summary, we have found a long-term association of CLOCK gene variants in the weight response after bariatric surgery and reproduced the association between rs3749474 variant and obesity, previously described. Carriers of the A allele of the rs1801260 variant and carriers of the T allele of rs3749474 variant were associated in our series with a worse weight response and a higher risk of MO, coinciding the risk alleles in the two phenotypes studied. 

## Figures and Tables

**Figure 1 nutrients-14-03472-f001:**
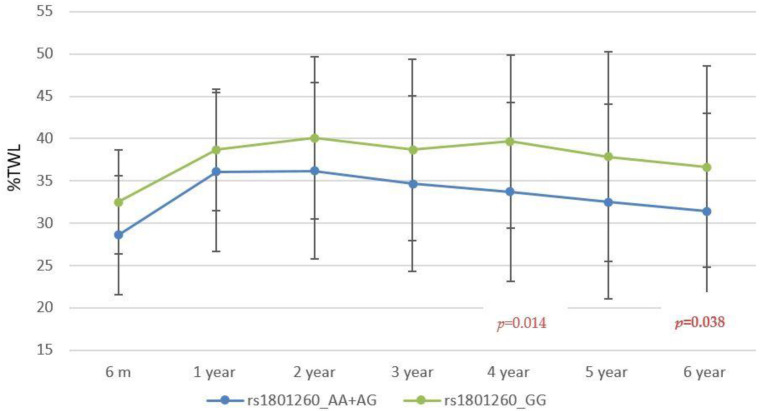
Mean %TWL in the follow up by rs1801260 genotypes.

**Figure 2 nutrients-14-03472-f002:**
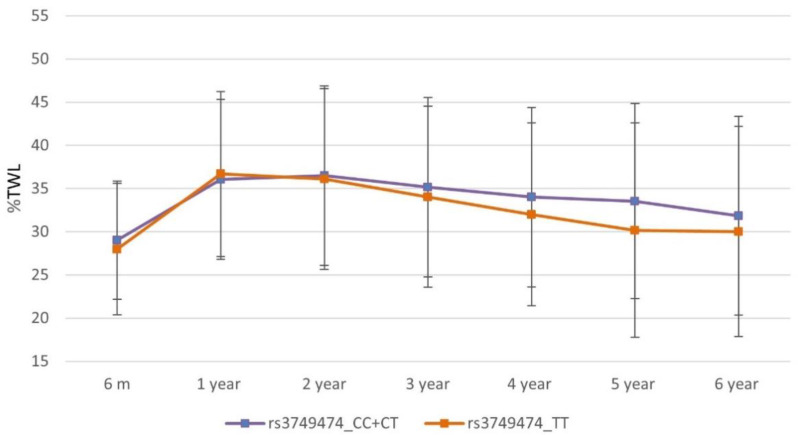
Mean %TWL in the follow up by rs3749474 genotypes.

**Figure 3 nutrients-14-03472-f003:**
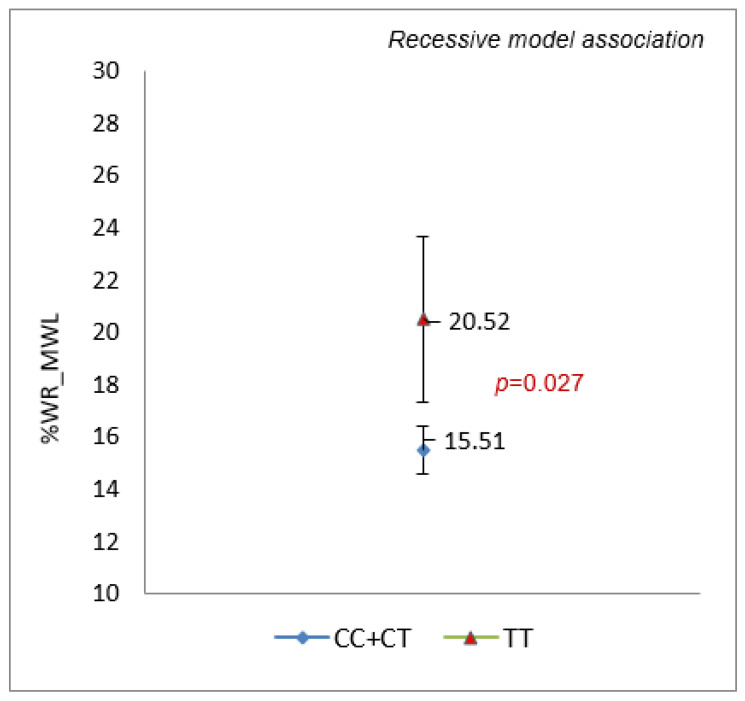
Association of rs3749474 genotypes with WR at year 6 in the malabsorptive cohort.

**Table 1 nutrients-14-03472-t001:** Absolute and relative frequency of genotypes of the included CLOCK gene variants.

SNP	Genotype	Controls n (%)	Cases n (%)
rs3749474	C/C *	100 (44.2%)	142 (37.9%)
	C/T	101 (44.7%)	152 (40.5%)
	T/T	25 (11.1%)	81 (21.6%)
rs1801260	A/A *	115 (50.2%)	202 (53.9%)
	A/G	90 (39.3%)	157 (41.9%)
	G/G	24 (10.5%)	16 (4.3%)
rs4580704	G/G *	30 (13.1%)	52 (13.9%)
	G/C	105 (45.9%)	169 (45.1%)
	C/C	94 (41%)	154 (41.1%)

* Reference genotype. SNP, single nucleotide polymorphism.

**Table 2 nutrients-14-03472-t002:** Profile of cases included in the genetic study (N = 375).

Variable	Value
Age, in years	44.79 ± 11.99
BMI_00, kg/m^2^	44.87 ± 6.59
Female gender, n (%)	259 (69)
T2D, n (%)	134 (35.7)
HTN, n (%)	182 (49)
%TWL_nadir,	38.79 ± 9.84
%EWL_nadir,	91.19 ± 23.69
%TWL_6y	31.67 ± 11.62
%EWL_6y	74.08 ± 26.89
%EWL > 50%, n (%)	311 (82.93)
%TWL > 20%, n (%)	317 (84.53)
%WR_MWL, median (IQR)	15.76 (7.99–28.69)

Mean (SD) unless otherwise stated. BMI_00, presurgery body mass index; T2D, type 2 diabetes; HTN, hypertension; TWL, total weight loss; EWL, excess weight loss; WR_MWL, weight regain from the maximum weight loss.

**Table 3 nutrients-14-03472-t003:** Mean values of weight response variables by genotypes. Global cohort (N = 375).

SNP	Genotype	n	%TWL_nadir	%TWL_6y	%WR_MWL
rs3749474	C/C	142	39.48 ± 0.86	32.26 ± 0.97	20.04 ± 1.55
	C/T	152	38.11 ± 0.78	31.45 ± 0.93	19.24 ± 1.31
	T/T	81	38.84 ± 1.07	31.03 ± 1.35	22.35 ± 2.41
rs1801260	A/A/	202	38.87 ± 0.67	31.22 ± 0.83	21.63 ± 1.36
	A/G	157	38.31 ± 0.82	31.73 ± 0.9	18.95 ± 1.37
	G/G	16	42.43 ± 2.31	36.7 ± 2.97	14.88 ± 3.61
rs4580704	G/G	52	39.09 ± 1.22	30.65 ± 1.5	20.39 ± 1.5
	G/C	169	39.48 ± 0.8	32.56 ± 0.9	19.29 ± 1.43
	C/C	154	37.93 ± 0.77	31.03 ± 0.95	22.73 ± 2.32

TWL, total weight loss; WR_MWL, weight regain from the maximum weight loss. Results adjusted by age, sex, T2D, HTN, type of surgical technique, and initial BMI.

## Data Availability

Not applicable.

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
