# Peer review of "Influence of CLOCK Gene Variants on Weight Response after Bariatric Surgery"

_nutrients, 2022, doi:10.3390/nu14173472_

Round 1
Reviewer 1 Report
The single-center retrospective study aimed at assessing association of single nucleotide polymorphisms of CLOCK gene with obesity and with long-term weight evolution after bariatric surgery (sleeve gastrectomy; Roux-en-Y gastric bypass; biliopancreatic diversion with or without duodenal switch; single anastomosis duodeno-ileal bypass with sleeve gastrectomy).
Adequate statistical analysis was performed.
The manuscript is well written and in a coherent manner.
The researchers found a long-term association of CLOCK gene variants with weight response after bariatric surgery. The variant rs1801260 seemed to have a protective effect for morbid obesity.
Author Response
Dear reviewer,
Thank you very much for your report about our submitted manuscript. We have updated the manuscript with the following issues:
-Format and minor spelling mistakes have been corrected
-Millar's symbols have been adapted to English
-The format of the references has been adapted to the requirements of MDPI. Thank you for your comments on the study, its objectives and the results observed. We hope that the attached version submitted will be considered suitable for publication in Nutrients.
Kind regards,
Dr Macarena Torrego on behalf of all authors

Reviewer 2 Report
Dear Authors
I have reviewed your paper with great interest.
I will accept your paper after a minimal revision.
My revision is:
Title: Very Good
Abstract: Very Good
Introduction and AIM: The problem and the aim are well descripting.
Results: Focus on and well described.
Discussion and Thread: effectiveness Focus ON.
Myalgia reflects generalized inflammation and cytokine response and can be the onset symptom of 36% of patients with high level of Interleukin-6 (IL-6) and tumor necrosis factor-α (TNF-α), please cite and discuss this paper:
Ripani U, Bisaccia M, Meccariello L. Dexamethasone and Nutraceutical Therapy Can Reduce the Myalgia Due to COVID-19 - a Systemic Review of the Active Substances that Can Reduce the Expression of Interlukin-6. Med Arch. 2022 Feb;76(1):66-71. doi: 10.5455/medarh.2022.76.66-71. PMID: 35422571; PMCID: PMC8976893.
References: Well chosen but to improve
Figures and Table: Very Good.
Author Response
Dear reviewer,
Thank you very much for your report about our submitted manuscript. We have updated the manuscript with the following issues:
-Format and minor spelling mistakes have been corrected
-Millar's symbols have been adapted to English
-The format of the references has been adapted to the requirements of MDPI.
Thank you for your comments on the different sections of the study (title, abstract, introduction and results). Regarding the discussion section and the suggestion “Myalgia reflects generalized inflammation and cytokine response and can be the onset symptom of 36% of patients with high level of Interleukin-6 (IL-6) and tumor necrosis factor-α (TNF-α), please cite and discuss this paper: Ripani U, Bisaccia M, Meccariello L. Dexamethasone and Nutraceutical Therapy Can Reduce the Myalgia Due to COVID-19 - a Systemic Review of the Active Substances that Can Reduce the Expression of Interlukin-6. Med Arch. 2022 Feb;76(1):66-71. doi: 10.5455/medarh.2022.76.66-71. PMID: 35422571; PMCID: PMC8976893.” we believe that it has probably been introduced by mistake as the content of the text is not directly related to the topic under study in our work. Regarding the reference section, the whole section has been updated according to MDPI style recommendations, also including the DOI in all references relating to scientific journal articles.
We hope that the attached version submitted will be considered suitable for publication in Nutrients.
Kind regards,
Dr Macarena Torrego on behalf of all authors
